



# Global distribution of hydrologic controls on forest growth

Casper T. J. Roebroek[1], Lieke A. Melsen[1], Anne J. Hoek van Dijke[1,2,3], Ying Fan[4], and Adriaan J. Teuling[1]

[1]Hydrology and Quantitative Water Management Group, Wageningen University & Research, Wageningen, the Netherlands
[2]Laboratory of Geo-Information Science and Remote Sensing, Wageningen University & Research, Wageningen, the Netherlands
[3]Environmental Sensing and Modelling, Environmental Research and Innovation Department, Luxembourg Institute of Science and Technology (LIST), Belvaux, Luxembourg
[4]Department of Earth and Planetary Sciences, Rutgers University, New Brunswick, NJ 08854, USA

**Correspondence:** Adriaan J. Teuling (ryan.teuling@wur.nl)

**Abstract.** Vegetation provides key ecosystem services and is an important component in the hydrological cycle. Traditionally, the global distribution of vegetation is explained through water availability by precipitation. Locally, however, groundwater can aid growth by providing an extra water source (e.g. oases) or hinder growth by presenting a barrier to root expansion (e.g. swamps). In this study we analysed the global correlation between precipitation, groundwater and forest growth, approximated
by the fraction of absorbed photosynthetically active radiation, and linked this to climate and landscape position. The results show that at the continental scale, precipitation is the main driver of forest productivity; wetter climates support higher energy absorption and consequentially more growth. But within all climates, landscape position substantially alters the growth patterns both positively and negatively. The influence of the landscape on vegetation growth varies over climate. The results display the importance of analysing vegetation growth in a climate-landscape continuum.

## 1 Introduction

Vegetation, key for many ecosystem services such as food production and climate stabilisation by absorbing $CO_2$ (Keenan and Williams, 2018), is an important component in the hydrological cycle. Water availability determines whether vegetation is present at all, while plants influence the local hydrological situation through interception of precipitation and transpiration of water absorbed in the root zone. Especially trees can impact the water fluxes substantially, returning significant amounts of
water back into the atmosphere (Kunert et al., 2017; Brauer et al., 2018). As a result, large scale changes in forest cover can influence continental-scale patterns of water availability and streamflow (Teuling et al., 2019). Because they can take up water from considerable depth with their extensive root systems (Canadell et al., 1996), trees are therefore highly adapted to the local climate and hydrologic regime (Wang-Erlandsson et al., 2016; Gao et al., 2014), making them resilient to weather anomalies, such as prolonged periods of drought (Nepstad et al., 1994; Kleidon and Heimann, 1998; Bowman and Prior, 2005; Walther
et al., 2019).

Plant available water, and with that vegetation growth, has traditionally been approximated by atmospheric states and fluxes such as precipitation (P) and evapotranspiration (ET). An example is the Köppen-Geiger climate classification, which links





ecosystems to the global distribution of precipitation and temperature (Beck et al., 2018). In line with this idea, Scheffer et al. (2018) recently showed that huge trees only occur in a climate niche with extensive amounts of rainfall. Local constraints

on vegetation growth have, with a similar reasoning, been approximated by the Budyko framework (Helman et al., 2017; Xu et al., 2013), which evaluates climate average precipitation, reference evapotranspiration and actual evapotranspiration to separate ecosystems into energy- or water-limited systems (Gunkel and Lange, 2017). Similarly, a recent study by Tao et al. (2016) showed a strong relation between tree growth and water yield (P - ET).

The distribution of atmospheric fluxes and states alone, however, can not fully explain vegetation growth worldwide (Fan,

2015). For example, oases appear as green islands in the middle of extensive arid regions, and gallery forests exist along the rivers in otherwise dry grassland areas under seasonally arid climates. In both cases lush vegetation can grow because the plant roots can tap into the groundwater to complement their water availability from local precipitation. The water table in these ecosystems is shallow in comparison with its surroundings due to topographic redistribution of precipitation surplus. Groundwater converges towards these niches, yielding relatively high water availability, decoupled from the local precipita-

tion (Fan, 2015). If the water table is shallow, precipitation can even become a hindrance for plant growth because it causes root-zone water-logging, limiting root oxygen uptake and hence limiting growth (Bartholomeus et al., 2008; Nosetto et al., 2009; Rodríguez-González et al., 2010; Florio et al., 2014). As such, land drainage conditions can alter the relation between precipitation and plant growth substantially, both positively and negatively.

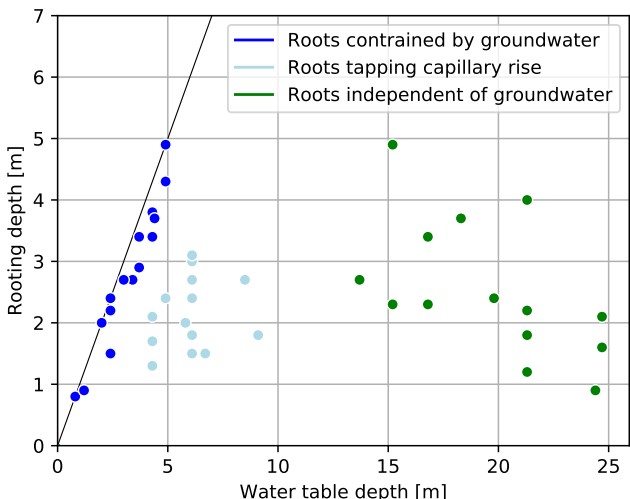

**Figure 1.** Illustration of the effect of water table depth on plant water uptake strategies, showing the rooting depth of 47 trees in Eastern Nebraska plotted against water table depth measured at their specific sites. Soil properties and precipitation are both relatively constant in the region. The roots can be divided in three distinct categories: (1) root growth is restricted by the groundwater, (2) roots are tapping the capillary rise, (3) roots are independent of the groundwater. Data from Sprackling and Read (1979) and interpretation adapted from Fan et al. (2017).





At the local scale, the effect of the water table on plant growth has been studied extensively. In an extensive case study, in an
area with similar soil properties and precipitation (Sprackling and Read, 1979), roots were found to fall in three categories (see
Figure 1): (1) roots terminating at or constrained by the groundwater, (2) roots tapping capillary rise and/or the groundwater
in the wet periods and (3) roots completely detached from the groundwater (Fan et al., 2017). At the farm scale, these patterns
were also observed (Zipper et al., 2015), with the conclusion that optimal plant growth occurs at the interface between the
groundwater limiting root respiration and roots being completely decoupled from the groundwater.

Site-based studies suggest that, at the landscape scale, rooting depth depends on the climate in the uplands, but on the water
table depth in the lowlands, presenting an optimal position where growth is aided by the groundwater while not suffering from
rooting space limitation (Zipper et al., 2015; Fan et al., 2017). At the global scale, Koirala et al. (2017) examined the influence
of the water table depth on vegetation growth. They found that both mechanisms, plant growth aided by groundwater in water
limited areas and plant growth hindered by groundwater due to oxygen stress, were reflected in the global satellite imagery
analysis. The questions that remain are what the interplay is between precipitation and groundwater for vegetation growth, how
landscape position determines this interplay over different climates, and how extensive the area is in which vegetation growth
is influenced by the groundwater.

Therefore, the purpose of this study is to understand and evaluate the global distribution of the effect of both precipitation
and land drainage (reflected by water table depth) on vegetation growth, and to assess the control of climate and landscape on
these processes. To do this, we make use of global high-resolution (30 arc-seconds) datasets of water table depth, precipitation
and tree growth, approximated by the fraction of absorbed photosynthetically active radiation (fAPAR). The relatively high
resolution for a global study allows us to account for landscape-scale features within computational limits (Fan et al., 2017).
We focus on trees, rather than vegetation in general, because they better represent the long term local hydrologic regime. At
the same time this lets us avoid confounding signals such as irrigation of annual crops, the response of annual vegetation to
seasonal availability of soil water and inter-annual variation. In this way we aim to evaluate plant productivity over a climate
gradient at the global scale, and quantify the global extent of vegetation growth influenced by the water table.

## 2 Materials and Methods

### 2.1 Input data

To approximate tree growth we used two different datasets. The first one is the MODIS fAPAR product, which is used as an
approximation of plant primary production (Wu et al., 2010). The data has a 15 arc-second spatial and an 8-day temporal reso-
lution (Myneni et al., 2015). For this study, we averaged the data over the period 2003 to 2018 and subsequently downsampled
it to a spatial resolution of 30 arc-seconds using bilinear interpolation (see Figure S1). The second dataset is a global map of
tree height, created from space-borne LIDAR images and validated against field measurements at different FLUXNET sites
(see Figure S2) (Simard et al., 2011). To solely focus on trees, the fAPAR dataset was filtered with the tree height data, using
a height threshold of 3 meters. For water table depth (WTD), the dataset by Fan et al. (2017) is used (updated version of the
original dataset in Fan et al. 2013). This dataset was produced by an integrated groundwater, soil water and plant root uptake



model at 30 arc-second resolution and at hourly time steps (see Figure S3). The precipitation data (WorldClim V2) was created by interpolating station observations using ancillary information, under which MODIS land surface temperature and a digital elevation model (Fick and Hijmans, 2017) (see Figure S4). A summary of the datasets is provided in Table 1. The time period
column describes the time frame of the input data of the specific studies to generate the datasets used here. It should be noted that both the WTD and fAPAR datasets were created using the MODIS MCD15A2H data and are therefore not completely independent. The MODIS data was used in the WTD model to describe the vegetation characteristics and to calculate the evapotranspiration and groundwater recharge fluxes. We believe this dependence to reflect the natural relation between vegetation and groundwater. Also, the impact on pixel-to-pixel correlations (between the fAPAR and WTD data) will be limited because
of spatial exchange of information in the WTD dataset, which causes the WTD to mainly reflect topography rather than local vegetation conditions.

**Table 1.** Summary of the datasets used in this study.

| Dataset | Spatial resolution [$arc-seconds$] | Time period | Version | Reference | Figure |
|---|---|---|---|---|---|
| fAPAR | 15 | 2003 - 2018 | MCD15A2H V6 | Myneni et al. (2015) | S1 |
| Tree height | 30 | 2005 | - | Simard et al. (2011) | S2 |
| Water table depth | 30 | 1961 - 1990 | V2 | Fan et al. (2017) | S3 |
| Precipitation | 30 | 1970 - 2000 | WorldClim V2 | Fick and Hijmans (2017) | S4 |
| Climate zones | 30 | 1980 - 2016 | V1 (present) | Beck et al. (2018) | S5 |
| Landscape classes | 30 | 1961 - 1990 | - | Text S1 | S6 |

## 2.2 Analysis procedure

To understand and visualise the relation between the hydrologic gradients and forest growth, the local Pearson correlation was calculated between (1) WTD and fAPAR and between (2) P and fAPAR. This was done by applying a moving window
($15 \times 15$ grid cells) to both datasets and correlating the values within that window. Windows containing less than 25 percent of the data were discarded. This approach was chosen over catchment binning, as used in previous studies (Koirala et al., 2017), to minimise compensation of contrasting relations (rooting space limitation in lowlands and groundwater convergence driven vegetation growth in uplands both occurring in a single catchment resulting in a net neutral relation between the water table and vegetation growth). Finally, each pixel contains a correlation value between the hydrologic gradient (WTD, P) and vegetation
growth. With this approach it is assumed that within each window ecosystems (e.g. forest age), soils (e.g. nutrient availability), management parameters (e.g. fertilisation), error in the input data and translation from fAPAR values to photosynthetic activity are homogeneous. The resulting correlation values are subsequently tested for significance, resulting in a negative, neutral





or positive category in each pixel. The threshold of significance was calculated by casting the correlation values into the t-distribution with Equation 1, in which $r$ corresponds to the correlation, $t$ to the t-value and $n$ to the number of samples.

$$t = r\sqrt{\frac{n-2}{1-r^2}} \tag{1}$$

This can be rewritten to calculate the correlation value based on the t-value:

$$r = \frac{t}{\sqrt{n-2+t^2}} \tag{2}$$

Using the percent point function of the t-distribution with a significance level of $p < 0.05$ and using a one-tailed approach, the significant t-value can be calculated. Feeding this value into Equation 2, the t-value can be translated into the threshold 100 correlation value. With 225 sample points ($15 \times 15$ pixel moving window approach, assuming all pixels contain values) this yields that the threshold for considering correlation significant is 0.11. In windows containing fewer data points, this threshold increases accordingly.

To investigate the interplay between P and WTD on forest growth, we combined the two significance maps, yielding nine distinctive classes (see Figure 2), henceforth called ecohydrological classes. This combination is visualised using a bivariate 105 colour scheme (Teuling et al., 2011). For the interpretation of the classes it needs to be considered that WTD is defined negatively; a higher value (less negative) corresponds with a shallower water table. Consequently a positive correlation between WTD and fAPAR means higher plant productivity with a shallower water table. A negative correlation signifies an increase in productivity for a deeper water table. A positive correlation between P and fAPAR means higher plant productivity with higher precipitation. To interpret the different classes, the key shown in Figure 2 is proposed, which is discussed in the next section. 110 The classes have been interpreted and named a priori, based on a review of literature (see Introduction) and the current state of understanding.

The effect of landscape and climate on the hydrologic controls of vegetation growth was characterised by analysing the obtained ecohydrological classes in different climate zones and landscape positions. A recent, high resolution Köppen-Geiger climate classification was used, based on the same precipitation data as used for this study (Beck et al., 2018)(Figure S5). 115 To asses landscape positions, we used a landscape classification based on the moving window mean and standard deviation of WTD ($5 \times 5$ pixels). Subsequently, the result was binned into 7 landscape classes: wetland and open water, lowland, undulating, hilly, low mountainous, mountainous, high mountainous (see Text S1 in the supplementary information). The classification scheme is depicted in Figure S7 and the resulting map is presented in Figure S6.

All maps are downsampled to a resolution of 5 arc-minutes by applying a majority kernel on categorical and a mean kernel 120 on continuous data. This was done to ease calculation and to be able to focus on the global patters. Some figures are displayed at their full resolution to discern finer patterns in the maps, in which case it is stated in the caption.





## 2.3 Ecohydrological classes

Based on the significance of the correlation analysis between WTD and fAPAR, and between P and fAPAR, we distinguish 9 ecohydrological classes. These are depicted in Figure 2. Below we provide a description of each class, discussing processes

that might play a role in the vegetation - hydrologic gradient relation, starting from the bottom left.

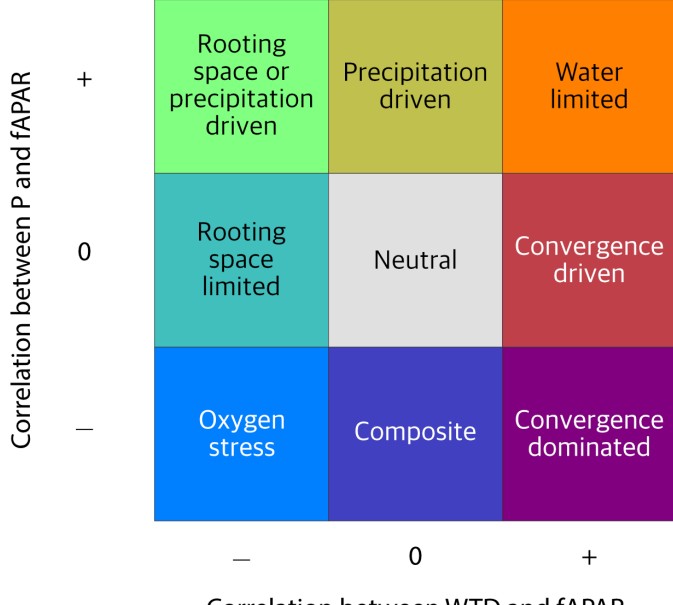

**Figure 2.** Ecohydrological classes and their interpretation of the combined spatial correlation maps between respectively WTD, P and fAPAR. The figure is used as the legend of Figures 3, 6 and 7

[*Oxygen stress*]; In this class, negative correlations with both hydrologic gradients suggests that plant growth is limited by higher precipitation and shallower groundwater, indicating an excess of water with poor drainage conditions. This combination causes root-zone water-logging, which limits root respiration (oxygen stress) and hence growth (Nosetto et al., 2009; Rodríguez-González et al., 2010; Florio et al., 2014; Zipper et al., 2015).

[*Rooting space limited*]; Here, plant growth is limited by shallower groundwater. In humid climates this indicates an excess of water in combination with poor drainage conditions. This class is largely similar to *Oxygen stress* except that there is no clear relation between precipitation and vegetation growth, which might be caused by the absence of a clear precipitation gradient. In arid and seasonally arid climates, the negative influence of the vicinity of the water table might be explained by high salt concentrations of the water in low landscape positions. Due to groundwater convergence, salts are transported to the lowest

positions in the landscape and high evapotranspiration increases the salt concentration dramatically, hindering plant growth (Jolly et al., 2008).



[*Rooting space or precipitation driven*]; This class is a transitional class between *Rooting space limited* and *Precipitation driven*. Either the negative correlation between WTD and fAPAR (rooting space limitation) or the positive correlation between precipitation and fAPAR (water limitation) explains the local tree growth gradients while the other correlation is caused by a

negative relation between WTD and precipitation. Often this negative relation can be explained by orography. Since WTD is roughly the inverse of altitude, locations with orographic precipitation (Fick and Hijmans, 2017) have a clear negative gradient between WTD and P. This negative correlation can sometimes also be explained by micro-climatic phenomena. This class can be interpreted as *Rooting space limited* if roots reach the groundwater and *Precipitation driven* if roots do not reach the groundwater. Alternatively, in the dryer parts of the world, this class can also be interpreted directly as forests growing on the

edges of basins where both a deeper water table and higher (orographic) precipitation help to counter growth limitation by high salt concentrations. In the centre of these basins the salt concentration is very high due to groundwater convergence transporting the salts and strong evapotranspiration. Higher rainfall in combination with well drained soils can flush away the salt, creating more favourable conditions. This explains both the negative correlation between WTD and fAPAR and the positive correlation between P and fAPAR.

[*Precipitation driven*]; Plant growth is enhanced by increasing precipitation and is decoupled from the groundwater table. This likely occurs in well-drained, upland positions, where roots cannot reach the groundwater, under climatic conditions where plant growth is slightly to severely limited by water availability. Here, precipitation is the main driver for productivity.

[*Water limited*]; Plant growth is stimulated by a shallower water table and higher precipitation, indicating a general lack of water. This likely occurs on mountain slopes where the water table is within root reach and in (semi-)arid climates where

plants depend on deeper ground water. In mountainous regions this class can, however, have another possible explanation. If the highest landscape positions receive less precipitation than the associated valleys, the relation between water table depth and precipitation will be positive. This can, for example, occur when the higher positions are above the zone of maximum precipitation (Miller, 1961; Junquas et al., 2016). Together, this means that higher vegetation growth can be expected in valley positions (higher vegetation growth with a shallower water table) with higher precipitation. In this case the class can be linked

to growth limitation at higher altitudes, like lower temperatures and a shorter growing season (Fan et al., 2009).

[*Convergence driven*]; Plant growth is stimulated by a shallower water table. This represents areas that receive water from surrounding, higher areas by lateral redistribution of the groundwater, as described in Fan (2015). This likely occurs in arid or seasonally arid climates where precipitation is low and irregular, but where the groundwater is within the reach of roots. These circumstances occur, for example, in desert oases and gallery forests (Fan, 2015). In mountainous regions this class can

also be related to different processes that are linked to higher altitudes (further from the water table generally means higher in the landscape), like lower temperatures (Leal et al., 2007), a shorter growing season (Fan et al., 2009) and lower nutrient availability (Leuschner et al., 2007), that hamper tree growth.

[*Convergence dominated*]; Plant growth is stimulated by a shallower water table but is limited by an increase in precipitation. This class corresponds to similar environments as described in *Convergence driven*. The negative correlation between precipi-

tation and fAPAR likely occurs in the following scenarios. Firstly, the orographic effect causes higher precipitation in regions with higher altitudes. In arid or seasonally arid regions this increase in precipitation is present but cannot provide vegetation





with as much water as the groundwater can in the lower positions, leading to an apparent negative correlation. Secondly, higher precipitation comes with higher cloud cover and lower temperatures, which in turn can limit plant growth. Here, vegetation growth mainly happens during the dry period, during which groundwater access is a major advantage for plant growth.

[*Composite*] This class displays no significant relation between the proximity of the groundwater and plant growth while plant growth is negatively influenced by precipitation. This can mean different things in different landscape positions. Firstly, in lowland regions it can be linked to oxygen stress (as in class *Oxygen stress*). If the water table is flat enough, no correlation is expected to be found but the negative correlation between precipitation and plant growth can still be present. Secondly, in mountainous regions absence of a correlation between water table depth and fAPAR can be linked to ecosystems being

completely detached from the groundwater. A negative influence of precipitation could be explained in different ways; (1) precipitation in convex positions in the landscape can lead to local over-saturation causing water-logging, decreasing plant growth, or (2) precipitation increases with altitude, and factors linked to a higher position in the landscape (as described in *Convergence driven*) are limiting plant growth.

    [*Neutral*]; This class contains the locations that show no significant correlation between either water table depth or precipi-

tation and fAPAR.

    Overall, there can be several process drivers in each ecohydrological class, dependent on climate and landscape position. In the next section, we will explore the global spatial distribution of the discussed ecoyhdrological classes.

## 3   Results

### 3.1   Global distribution of ecohydrological classes

Figure 3 displays the global distribution of the ecohydrological classes that were described in the previous section. In more than half of the pixels, forest growth is significantly influenced by the water table depth, and in more than 80 percent by precipitation, confirming the hypothesis that P is an important but not the only driver of forest growth. All different classes are present in this global analysis; to a varying degree on all continents and in all climate zones. Clear cases of water limitation (both correlations positive) are relatively under-represented as most water limited areas were filtered out by applying a tree

height threshold of 3 meters. The results show that the water table depth plays a major role in determining forest growth, even in regions that are traditionally seen as energy limited environments. WTD clearly shows a different signal than P, since the correlation between the two gradients can both be strongly positive (more precipitation with a shallower water table) or negative (more precipitation with a deeper water table, likely caused by orography) (see Figure S10).

    Four insets (15 degrees) are displayed in Figure 3. The same insets are displayed in Figure S13 to Figure S16 together with

the input and individual correlation data. Inset *A* (Figure S13) shows the Mississippi river valley on the left and the southern part of the American East Coast on the right. The river valley itself shows a neutral or negative correlation between both WTD and P with fAPAR, representing an environment where too much water leads to over-saturation and water-logging which hampers tree growth. This corresponds to the ecohydrological classes *Oxygen stress* and *Rooting space limited*. Further away from the river, the relation between P and fAPAR changes to positive leading to a classification of *Rooting space or precipitation driven*,





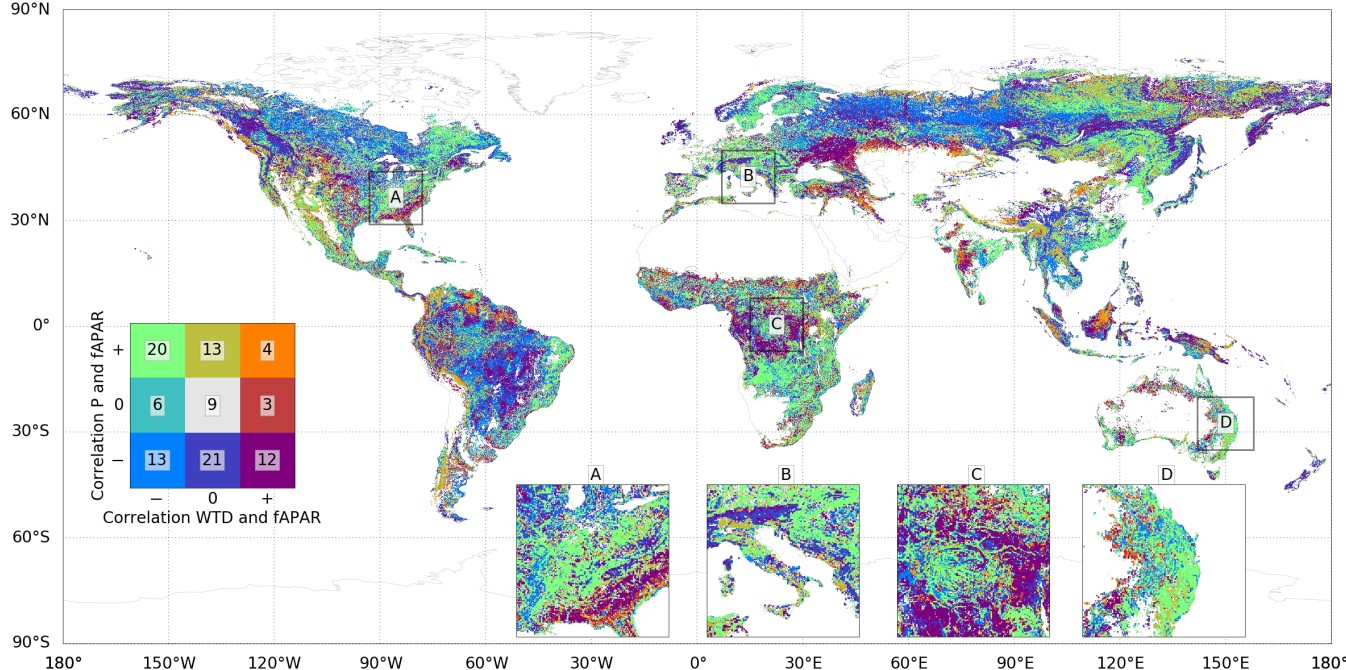

**Figure 3.** Global distribution of ecohydrological classes. The legend indicates the percentage of grid cells in the different classes. The map is downsampled to a resolution of 5 arc-minutes. For a bigger version of the map see Figure S11 and for the version at original resolution of 30 arc-seconds see Figure S12. Note that the percentages add up to 101, which is caused by rounding.

which links a higher position in the landscape to more precipitation and more vegetation growth. Towards the coast, on the interface between Georgia, Alabama and Florida, forest growth is *Convergence dominated* and in some places *Water limited* and *Convergence driven*.

Inset *B* (Figure S14) shows South-Eastern Europe with the Alps. In this mountainous region, plant growth is predominantly detached from groundwater influences (hardly any significant correlations between WTD and fAPAR). In the southern part

of the Alps, forest growth is precipitation driven while the northern part falls in the *Composite* class, featuring a negative correlation between P and fAPAR. In mountainous regions this class corresponds to an ecosystem that is detached from the groundwater and grows best in the lower or mid landscape positions. Higher up in the mountains, vegetation growth is disturbed by factors such as low temperatures, shallow soils and a reduced growing season. The hilly regions around the Alps are predominantly classified as *Rooting space or precipitation driven*, as in inset *A*. This corresponds to enhanced tree growth in

the higher locations, associated with more rain and more rooting space. Another interesting feature in this inset is the Pannonian Basin (north-east in the inset), showing a similar pattern as the Mississippi valley of *Rooting space limited* vegetation growth. Groundwater convergence from the surrounding higher regions causes very shallow water table depths in this area, hampering forest growth.





Inset *C* (Figure S15) depicts the Congo river basin. The Congo river and its side-channels show similar patterns of increased
vegetation growth on levees, leading to a *Rooting space or precipitation driven* classification. The regions to the south and east
of the Congo river basin are dominated by savannas. These savannas receive a substantial amount of precipitation yearly, but
rainfall is not evenly distributed over the year and makes water relatively scarce in comparison with the energy input at these
latitudes (Verhegghen et al., 2012), leading to a classification of *Convergence dominated*. Areas at high altitude in this closeup
shows a *Composite* class; most forest growth occurs at the foot of mountains or on the slopes, while higher locations are less
suitable due to lower temperatures and a shorter growing season.

Inset *D* (Figure S16) shows an orographic region in Eastern Australia, where vegetation growth is driven by the precipitation
gradient. The lowland west of the mountain range (Great Dividing Range) is classified as vegetation limited by *Rooting space
limited*. Converging water from the mountain range causes a shallow water table depth in this region hampering forest growth.
The most western part of this inset that still contains trees receives between 250 and 500 mm precipitation per year. This region
is *Convergence driven*, also receiving water from the higher areas.

All four insets display a high spatial variability in ecohydrological classes, demonstrating that the local interplay in climate
and landscape position highly influence which hydrologic driver stimulates or hampers forest growth.

## 3.2 Local examples at high resolution

To better visualise and understand the patterns of ecohydrological classes, detailed maps of the input, correlation and output
maps are displayed in Figure 4 and Figure 5. Landscape position is approximated and displayed based on the standard deviation
of the WTD map (which is the main constituent of the landscape classification procedure). This representation was chosen over
the landscape classes, used throughout the rest of the paper, to obtained a more detailed visualisation.

The presented patterns in Figure 4, displaying the western Amazon, show a clear overlap with ecosystem functioning as
described in Ferreira-Ferreira et al. (2014). The river and its major contributing streams display the *Rooting space or precipi-
tation driven* class. Considering the (slightly) negative correlation between WTD and P, this can be attributed to rooting space
limited growth: the vegetation on the natural levees next to the channels are known for the highest and most diverse forests
of the Amazon (High Varzea in Ferreira-Ferreira et al. (2014)). On these levees the trees have more rooting space, receive
more precipitation and suffer comparatively little from the inundation that characterises these rivers, leading to optimal growth
conditions. In the depressions between streams (especially on the eastern side of these maps), forest growth is classified as
*Oxygen stress*. Here forests suffer from the very frequent inundations that hampers their respiration. These same areas feature
a positive relation between P and WTD, linking precipitation to percolation and a higher groundwater table.

The western part of the maps show *Convergence dominated* forest growth. This area is higher than the eastern part, present-
ing fewer streams, and has a (slightly) higher relief, making inundation much more rare. This area agrees with the mapping
of the White-sand Ecosystems as published by Adeney et al. (2016). These ecosystems have sandy, very well draining soils.
Even slightly elevated surfaces know temporary periods of draught with lower vegetation growth. Tree growth at the lowest
positions in these landscapes is higher, causing the *Convergence dominated* classification. In the hilly, north-eastern part of

**Figure 4.** High-resolution illustration of ecohydrological classification in the Amazon. Input and correlation maps are shown at full resolution of 30 arc-seconds. Note that the white pixels in the upper left map (ecohydrological classes) represent the locations where the correlations were not calculated due to the tree height falling below the threshold value of 3 meters.





maps forest growth is also classified as *Convergence dominated* as well as *Water limited* which is in stark contrast with the general perception of water abundance for vegetation growth in the Amazon region. This can be explained by the high amount
of available energy, even with respect to such extensive amounts of rainfall. At the foot of these hilly regions vegetation can reach the groundwater, and consequentially grow faster, thus causing a *Convergence dominated* classification. If the vegetation in a whole window cannot reach the groundwater anymore this turns into the *Water limited* class.

The second high resolution example (Figure 5) shows India. The western part of India features a mountain range (Western
Ghats), which is a strong orographic zone, receiving moisture from the Arabian Sea (especially during the monsoon season). This zone is predominantly classified as *Rooting space or precipitation driven*. In contrast with the Amazon example, this class is caused here by the precipitation driven vegetation (positive correlation P and fAPAR), as the groundwater is too deep to be reached by the vegetation. The negative correlation between the WTD and fAPAR is caused by the strong orographic gradient, with higher precipitation in higher areas (with a lower water table). This negative gradient can be seen in the lower right subplot
of Figure 5.

Behind the mountain range lies a vast rain shadow, receiving very little precipitation (Climate classes BWh and BSh). This area can be subdivided in two different zones; a southern and northern zone. Although they receive similar yearly amounts of precipitation the northern zone contains much more forests than the southern zone (which is mainly filtered out in this analysis since vegetation height is mostly under the threshold value of 3 meters). This stark difference can be attributed to the distance
of the water table to the surface. As can be seen in the upper right subplot of Figures 5, the southern zone has much deeper groundwater than the northern zone. The forest growth classification in the northern zone, following the same rational, is *Convergence driven*, *Convergence dominated* and *Water limited*; forest growth is highest in the lowest landscape positions with the easiest access to the groundwater as additional water source. Further east the amount of precipitation rises again (around 81°E and 18°N). This area features higher topography but a relatively shallow water table (plateau). This combination causes tree
roots to be constrained leading to the *Oxygen stress* classification. In contrast, the Eastern Ghats (first mountain range of India seen from the Bay of Bengal) show ecohydrological classes *Rooting space or precipitation driven* and *Composite*, which are linked to the orographic effect and decrease in temperature and growing season at higher altitudes.

When zooming in even further on the Amazon basin (see Figure S17) and India (see Figure S18), the potential of this high
resolution analysis becomes apparent. In Figure S17 individual levees and gullies can be identified based on the ecohydrological classes, demonstrating local differences in water availability for forest growth. In Figure S18 the strong gradients of the orographic effect and the driving effect of groundwater proximity as alternative water source can be observed.

### 3.3 Landscape and climate as drivers of the hydrological controls

To characterize the influence of landscape and climate on the governing processes, the data have been segregated on Köppen-
Geiger climate classes and landscape position classes. The results for four major climates are shown in Figure 6. Figure 6a shows clear patterns in both landscape positions and climates. The arid climate (BWh) has much lower fAPAR values than the



**Figure 5.** High-resolution illustration of ecohydrological classification over India. Input and correlation maps are shown at full resolution of 30 arc-seconds. Note that the white pixels in the upper left map (ecohydrological classes) represent the locations where the correlations were not calculated due to the tree height falling below the threshold value of 3 meters.





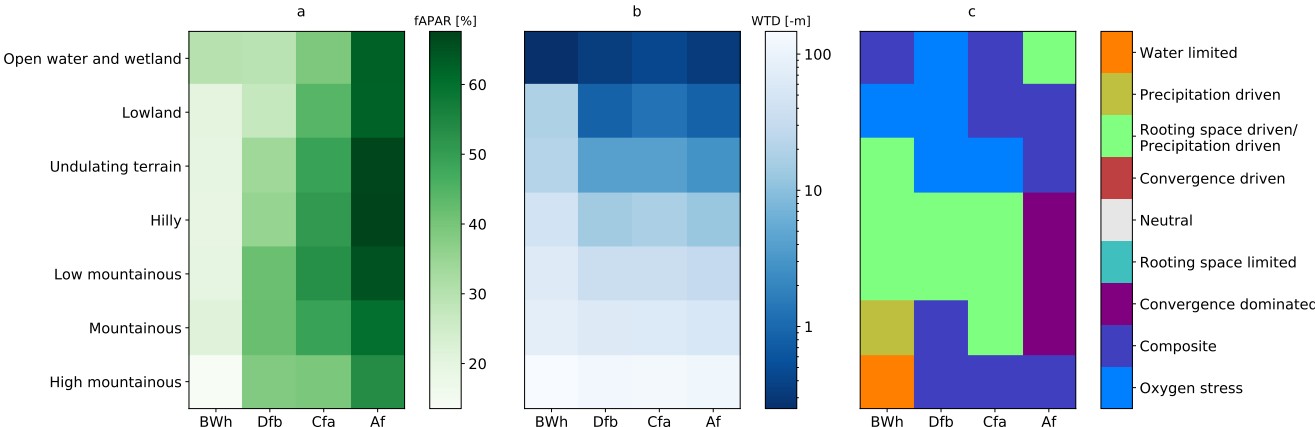

**Figure 6.** Distribution of average ecohydrological functioning as a function of landscape position and climate. This Figure shows a subset of the Köppen-Geiger climates for clarity; namely arid (BWh), boreal (Dfb), temperate (Cfa) and tropical (Af). For an extended version containing all the climates see Figures S19, S20 and S21 in the supplementary material. (a) mean fAPAR, (b) mean water table depth and (c) prevalent ecohydrological class (after removing the cells in the neutral class).

tropical climate (Af) and the intermediate temperate (Cfa) and boreal (Dfb) climate fall in between, confirming the hypothesis that tree growth, at climate scale, follows the gradient of precipitation. Both extremes in the landscape (High mountainous and Wetland) display lower fAPAR, except for the arid climate in which the lowest position in the landscape corresponds to

the highest fAPAR. The highest fAPAR in the other climates falls in the intermediate landscape positions. Figure 6b shows mean water table depth in the different climate and landscape positions. As expected, the water table is generally deeper in arid climates compared to wetter climates in similar landscape positions, except for the lowest landscape position.

The ecohydrological classes (Figure 6c) show a consistent pattern. In the lowest positions in the landscape, vegetation growth is limited by rooting space. This is followed by a region that is driven by the precipitation gradient (*Rooting space or precipita-*

*tion driven* and *Precipitation driven*). *Rooting space or precipitation driven* displays a negative correlation between WTD and fAPAR here, as a consequence of more (orographic) precipitation at higher locations in the landscape. This process is similar in most climate zones (see Figure 6c and Figure S21), but the threshold within the landscape is lower in arid environments, following a general lower water table depth at similar landscape positions (Figure 6b). Exceptions are the tropical climates (Af and Am), in which vegetation growth in mid-landscape positions is driven by groundwater convergence, hinting at a relative

scarcity of water in comparison to the energy availability.

In the temperate and tropical climates, where precipitation is generally high, limited rooting space in the lowest landscape positions suppresses growth. Consequentially, the optimum in fAPAR occurs higher up in the landscape, where rooting space is no longer a limitation. In the arid and boreal climates the lowest position in the landscape is favourable (higher fAPAR than adjacent landscape positions). In arid climates this is associated to groundwater convergence from large areas as water

availability from precipitation is generally low. In a boreal climate this optimum might instead be caused by higher temperatures





at lower landscape positions. The highest landscape positions are decoupled from the groundwater and are therefore classified as *Composite*, *Precipitation driven* and *Water limited*. The *Composite* class is in this context associated with a growth optimum on the slopes, while valleys and ridges show a reduction in plant primary production. Not surprisingly the highest positions in the arid climate has the lowest fAPAR of all positions and is classified as *Water limited*.

### 310  3.4   A novel framework to link forest growth to the hydrologic gradients in a climate-landscape continuum

Based on our results, we propose a framework for tree growth in different landscape positions and climates, displayed in Figure 7. In arid regions the vegetation is concentrated in the lowest landscape positions which correspond to the notion that vegetation in deserts predominantly thrives in oases, which are driven by groundwater convergence of extended areas. Another optimum, though with lower tree growth, exists higher up in the landscape, where the mountains are wetter, cooler and greener
than the surrounding desert basins (better visible in Figure 6a).

In the temperate and tropical climate, only one growth optimum is discernible. In the tropical climate this optimum corresponds with the region driven by local groundwater convergence (see Figure 6a and c). This optimum lies exactly on the point where the correlation between water table depth and fAPAR switches from positive to negative, implying the existence of a distance to the groundwater that is shallow enough to be accessible for roots and deep enough for it not to negatively influence
root growth. In the temperate climate the optimum of vegetation growth lies in the zone classified as *Rooting space or precipitation driven*, with a negative correlation between WTD and fAPAR. In comparison with lower positions in the landscape, this zone displays a positive correlation with precipitation, hinting at precipitation driven vegetation, only displaying a negative correlation between WTD and fAPAR because higher precipitation falls at higher locations. This suggests that vegetation is detached from the groundwater in these mid-landscape positions, with vegetation growth being limited by water availability.
In the lowest landscape positions even more water is available but, because the shallow groundwater confines the root zone, plants can not take optimal advantage of the resource.

The boreal climate shows a very similar pattern as the temperate climate, although fAPAR values are lower. This climate does show a second optimum in fAPAR in the lowest landscape position (better visible in Figure 6a), similar to the arid climate.

## 4   Discussion

### 330  4.1   Correlation in hydrologic gradients

The presented results show that global gradients of P and WTD have a substantial effect on forest growth. These gradients, however, are not independent, which needs to be considered when interpreting the results. The correlation between precipitation and WTD is shown in Figure S10 and shows clear spatial patterns of both positive and negative values. A negative correlation corresponds to higher precipitation with a deeper water table while a positive correlation indicates lower precipitation with
a deeper water table. In terms of processes, these relations can best be explained when considering that water table depth is roughly the inverse of altitude (especially in hilly and mountainous terrain). A negative correlation between WTD and P would







**Figure 7.** Conceptual framework summarizing the links between fAPAR, water table depth, the correlations and implications for the patterns of rooting depth across climate and landscape classes. Different percentages of fAPAR are depicted as tree symbols, the ecohydrological classes are shown as arrows.





correspond to more precipitation higher in the landscape, which is linked to orographic precipitation. Positive correlation values between WTD and P seem to often occur in either low-lying areas, where more precipitation yields more percolation and a shallower water table, or in mountainous areas, which could correspond to a decrease in precipitation with altitude due to a

loss of atmospheric moisture due to orographic precipitation in lower lying areas. These processes are clearly present in the class *Rooting space or precipitation driven*, but a correlation between P and WTD should be considered in all other classes as well. For example, Borneo is partly classified as *Water limited*. As a tropical mountainous island, *Precipitation driven* would be expected. Considering the a positive correlation between WTD and P (higher precipitation at lower locations in the landscape), the term *Water limited* that was assigned to this class does not correctly describe all processes in this region.

## 4.2    Variation over time

In this study we analysed forest growth under long term average gradients of water table depth and precipitation, even though both hydrologic gradients can show considerable seasonality. We acknowledge that seasonality in precipitation and water table depth can influence the local vegetation type, but we believe that by focusing on forests only, long term averages in hydrologic gradients can provide useful insights. It can be assumed that forests are strongly adapted to the local hydrological regime and

therefore mainly respond to long term changes in these regimes. This approach was chosen to understand the global patterns of long term ecosystem behaviour and water resources. By using the long term average gradients we focus on the question if and where forests are driven by the groundwater, precipitation, or both.

### 4.3    A start for a more sophisticated forest growth representation in global modelling studies

Many global Earth system modelling studies do not account for water table depth as a driver of forest growth. Our results sug-

gest that landscape-scale interaction between vegetation and groundwater, including lateral convergence, moisture and oxygen stress, is important in most parts of the world and should be better represented in these Earth system models. Groundwater can either be an extra water source for vegetation growth, but also a constraint on root growth and with that vegetation growth. The presented framework can serve as a first approach to account for both forest growth stimulation and growth limitation based on precipitation and water table depth in a climate-landscape continuum. Local examples, such as the the Amazon river

and the mainland of India, show a consistent overlap between the presented patterns and expected tree growth, based on the understanding of the ecosystems. It needs to be considered that seasonality and inter-annual variability of both precipitation and the water table can change the presented patterns substantially, but the understanding of average ecosystem behaviour on a climate-landscape continuum can be used as a baseline in further studies. The global importance of the landscape-scale water table variability on forest growth proves that it needs to be considered in global environmental modelling.

## 5    Conclusions

The goal of this study was to relate two hydrologic gradients (precipitation and water table depth) to forest growth on a global scale. The presented results show that across most of Earth's surface, water is an important control on plant productivity,

determining the presence of vegetation and constraining it's growth. Water table depth, an often ignored parameter in global Earth system modelling, displays a significant influence on vegetation growth in more than 50 percent in the forested pixels,

both positively (e.g. tree growth stimulation in oases) and negatively (e.g. tree growth hindrance in swamps). In a substantial part of the globe, this influence does not overlap with an influence of precipitation, although both gradients generally show strongly spatial correlation.

Inter-climate analysis demonstrates that, at the continental scale, vegetation growth is strongly driven by precipitation; vegetation in wetter climates shows higher energy absorption. Within these climate zones, vegetation growth can substantially

change over the landscape gradient. The effect of landscape is, however, not constant in all climate zones. As hypothesised, vegetation growth in arid regions is mainly driven by groundwater convergence, showing the highest energy absorption in the lowest landscape positions. In more humid climate zones, tree growth presents an optimum in mid-landscape positions. Below this optimum a shallow ground water table limits root growth and vegetation development, while at and above this optimum tree growth mainly follows the precipitation gradient. The proposed framework illustrates the importance of coupling landscape

and climate together to describe vegetation patterns world wide, tying root growth and water availability from precipitation and groundwater together. In the light of global changes in hydrologic gradients and land use, the water cycle will substantially change in the future. To predict the changes and mitigate the effects, water availability and root growth should be considered in global environmental modelling.

*Author contributions.*  CTJR designed and carried out the research and analysis under supervision of AJT, AHvD and LAM. YR helped with

the interpretation of the results. All the authors contributed to the writing of the manuscript.

*Competing interests.*  We declare that there are no competing interests.

*Acknowledgements.*  The input data used for this study can be found through the references provided in Table S1 in the supplementary information. The correlation maps, landscape classification map and the map of ecohydrological classes (both original and downsampled resolution) can be found at: https://www.hydroshare.org/resource/38ac7dd90c7d4353bb492604981782f0/. I would like to thank Agnese

Orzes, Bram Droppers and the co-authors for countless discussions and feedback on the methodology, interpretations and final text. Icons in Figure 7 were adapted from Borner et al. (2010).





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
