# Peer review of "Global distribution of hydrologic controls on forest growth"

_Hydrology and Earth System Sciences, 2020_

## Referee Comment (RC1) · Anonymous Referee #1 · 25 Mar 2020

In this interesting work, the authors classify the land surface into patterns of dominant effects on forest growth by precipitation or water table depth (wtd) using satellite imagery of fPAR as a proxy for forest growth, modelled wtd, and measured and interpolated precipitation. The analysis is based on spatial correlations between long term averages of the high resolution data sets. They find that the relationship between precipitation and fPAR is prevalent, but the effects of wtd are still widespread and important. The authors also illustrate variations of the relationships as the result of local climate conditions and landscape characteristics. As the authors convincingly explain in a paragraph in the discussion, the results prove that in current modelling approaches of the land surface using exclusively precipitation as a hydrologic control on forest growth is not sufficient and the work is therefore timely and relevant. The

paper is very well written, the analysis steps are clearly explained including underlying assumptions and the results are logically structured and interpreted. I was wondering, however, why the authors used full correlations all the way through their analysis when the scope was to actually isolate the hydrological control/ contribution. As mentioned in the description of several ecohydrological classes and sometimes in the interpretation of the results, spatial covariates like temperature play a role and will explain some of the patterns of correlations found, especially those with precipitation. So, why not remove at least the contribution of spatial gradients in temperature as a known important control on forest growth by partial correlations to narrow down the contributions of the hydrological controls? I might pose a similar question regarding the relationship between precip and wtd, which might also be split more rigorously. However, the authors take this into account in the interpretation and explain well in the paper, so I do not pose this a major point of discussion. Overall, the work the authors present in their paper is scientifically interesting and relevant, methodologically mostly logical (next to the one major point stated above, I pose some minor methodological questions below that need clarification or justification in my opinion), and is presented in an excellent way regarding both text and figures. I see the need for revision and minor clarifications before publication.

Minor aspects that need clarification/ discussion and potentially changes in the manuscript:

- Consistency of the long-term averages of the data sets: As shown in table 1 of the main text, the length and the periods that they represent differ by 10 years and more between individual data sets. How might this affect the consistency of the long-term averages that are the basis of the analysis? Secondly, the data availability of at least the fPAR dataset will vary seasonally due to snow or cloud effects. Has this issue been considered and taken into account in some way in order to prevent the longterm averages to be seasonally biased?

- Is the scope to analyse hydrological control of trees or of forests? From the title I
expected only forests, but basically all results are based on any pixels having a canopy height>3m independent of any definition of a forest, eg tree density. The assumption that the 'translation from fAPAR values to photosynthetic activity are homogeneous' (l. 91) in each moving window appears strong when only the threshold of 3m is used as a filter criterion and in reality several vegetation types might be mixed in the pixel. A slight rewording in the first and a clarification in the second case are appreciated.

- Are only those correlations displayed and evaluated that were tested as significant? Have you tried whether the results strongly change of you apply other criteria in addition, such as a (higher) correlation threshold? A threshold of 0.11 for a significant correlation for fully available spatial windows (l.101) is quite low as to have a strong meaning for the interpretation.

---

## Short Comment (SC1) · 1 Apr 2020

**Answer on Anonymous Referee #1**

Dear Reviewer,

Thank you for your review and your enthusiasm about the study. Below we address the comments you made on the paper.

Main points:

*"As mentioned in the description of several ecohydrological classes and sometimes in the interpretation of the results, spatial covariates like temperature play a role and will explain some of the patterns of correlations found, especially those with precipitation."*

As the reviewer correctly describes, other covariates will play a role in the relation on vegetation growth (approximated with fPAR). In this study we chose to focus on water table depth and precipitation. By using the method of windowed correlation, we are looking at the growth from a local scale perspective, assuming the covariates to be homogenous within each window. In practice this assumption will not always hold, which will decrease the explanative power of WTD and P on fPAR and most likely yield lower correlation values. We are testing for the significance of the correlation values (with relatively low thresholds, more on that later). Therefore, we isolate the effect of WTD and P on the vegetation growth. The covariate that the reviewer specifically mentions is temperature, but vegetation age, species, soil type, salinity and slope aspect would in some cases be equally important.  By looking specifically at temperature, it will be quite hard to separate its effect from that of precipitation and the water table depth directly (see Figure 1 and 2).  Especially WTD and temperature show a very consistent pattern of correlation values. Without including temperature in the descriptions of the classes however, vegetation growth patterns in especially mountainous areas are hard to express. Especially in these areas P and T correlate very well. If you find these maps useful and illustrative we can add them to the discussion section of the paper.

[Figure]

*Figure 1: Correlation between P and T, created similarly to the correlation maps in the paper.*

[Figure]

*Figure 2: Correlation between WTD and T.*

Minor points:

- *"Is the scope to analyse hydrological control of trees or of forests? From the title I expected only forests, but basically all results are based on any pixels having a canopy height>3m independent of any definition of a forest, eg tree density."*

We are interested in all natural vegetation but applied a filter of 3 meters to exclude most farmland as the signal of the natural controls will be heavily distorted. This threshold to classify trees seemed reasonable and is often used in literature. A pixel containing vegetation with an average height of 3 meters is consequently called forest. For clarity we will explain the use of 'forest' better in the methodology.

- *"Consistency of the long-term averages of the data sets: As shown in table 1 of the main text, the length and the periods that they represent differ by 10 years and more between individual data sets. How might this affect the consistency of the long-term averages that are the basis of the analysis?"*

In this study we are using data with high resolution, obtained from long term observations. The period mismatch will presumably lower the correlation values somewhat, but significance might still be tested. As the windowed correlation approach looks at local scale patterns, the mismatch will presumably only change the absolute values (of P and WTD), but the gradient within each correlation window will likely remain equal, as the landscape forms are still the same. This will not always work, in such cases as local water extractions being implemented, but on average on the whole world these changes will be relatively minor. We believe this mismatch will not substantially influence the conclusions.

- *"the data availability of at least the fPAR dataset will vary seasonally due to snow or cloud effects. Has this issue been considered and taken into account in some way in order to prevent the longterm averages to be seasonally biased?"*

The following fPAR data source has been used: https://developers.google.com/earth-engine/datasets/catalog/MODIS_006_MCD15A3H. It uses the best value for each pixel in four consecutive days. The long term averaging has been performed on this data directly to mostly avoid phenomena such as cloud effects.

The biases in the averaging and the seasonal cycle are addressed by the regional scale on which the correlations are calculated. Within each 15x15 grid cell window these biases are assumed homogeneous. The absolute values will shift, but the gradients would remain equal, yielding similar correlation values.

- *"The assumption that the 'translation from fAPAR values to photosynthetic activity are homogeneous' (l. 91) in each moving window appears strong when only the threshold of 3m is used as a filter criterion and in reality several vegetation types might be mixed in the pixel."*

This is indeed a good point. We assume not only vegetation type, but also vegetation age to be homogeneously distributed over each window (which would then make it justifiable to also assume a similar conversion function between fPAR and actual photosynthesis). Locally this assumption will not always hold but we believe this assumption is reasonable for a global synthesis and that the errors the assumption induces not to substantially alter the final observations and conclusions.

- *"Are only those correlations displayed and evaluated that were tested as significant? Have you tried whether the results strongly change of you apply other criteria in addition, such as a (higher) correlation threshold? A threshold of 0.11 for a significant correlation for fully available spatial windows (l.101) is quite low as to have a strong meaning for the interpretation."*

Currently we are using a alpha threshold value of 5%, which yields a significant correlation value (for n=225) of 0.11. We believe this is the most straightforward way of interpreting the data, as we are not interested in the explanatory power of WTD and P but merely in its significance in driving vegetation growth.ß

---

## Referee Comment (RC2) · Anonymous Referee #2 · 18 Apr 2020

General comments: In their work the authors present a method where they link forest growth patterns to precipitation and water table depth on the global scale. They use long term high resolution satellite products for fAPAR, modelled groundwater table depth and globally distributed precipitation. They have developed a classification scheme of ecohydrological classes based on the correlation between water table depth and fAPAR as well as precipitation and fAPAR. To assess the impact of climate and landscape position on the distribution of these ecohydrological classes, the authors make use of the Köppen-Geiger classification (climate) and 7 landscape classes derived from the global water table depth map. They discuss and illustrate their findings for several regions of the globe. In the end, based on their findings they develop a conceptual framework of forest growth and its link to hydrologic gradients in the landscape.

Finally, the authors discuss how their findings can support a better representation of forest growth in global environmental modelling which is still a relevant question. The manuscript is well written and conceivable. It is provided with an extensive supplement which contributes to the understanding of the manuscript. There are some small points which should be clarified because they might lead to misinterpretations which, I will address in specific comments section and section for technical correction. However, I have two major points which should be discussed by the authors with more emphasis. How have the landscape positions been validated, the map presented in Figure S6 makes sense at the global scale, but how valid are the results if you look at the landscape scale, where the authors develop their conceptual framework? I assume this can be easily done with global topography data such as SRTM. I would encourage the authors to discuss this a little bit more in detail since the landscape position is a critical part in your analysis. In the description of the Ecohydrological classes in section 2.3 I would stronger present the effects of temperature on forest growth in the higher landscape positions to avoid misinterpretations. Since this class is mostly present in the higher landscape classes of the temperate regions.

Specific comments

statement line 12 to 14 In my mind this statement is only true for water limited areas. For more humid, energy limited environments like the temperate and boreal zones I am not sure whether water availability determines whether vegetation grows or not especially when it comes to trees. I would argue that in the colder climates and higher mountainous areas plant growth an especially Tree growth is also limited by temperature which can be clearly also seen by the tree line distribution in the high mountain areas in the temperate regions as well as in the northern climates.

Statement line 16 to 20 This statement might be true on large continental scale, however as experiences of the drought years 2018 and 2019 in Europe have shown that forests mainly consisting of species trees species with shallow roots such as spruces suffered serious damages during the droughts.

[Figure]

Statement line 45 to 47 Rooting depth also depends soil properties like the existence of a layer of higher density in the soil profile. This is for instance very often the case in landscapes which have developed after the glaciation period or have been influenced by glaciation (e.g. in North America, Central Europa, Northern Part of Asia).

Figure 6: I would have expected a stronger temperature effect on forest growth also in the lower landscape classes like low mountain areas and hilly landscapes. How can this be explained?

Figure 7: For the boreal and temperate regions the figure indicates a deep and unchanging rooting depth from low mountainous, mountainous and high mountainous regions. This is misleading. In fact in these areas the rooting depth decreases with elevation. In the higher elevations only shallow soils over bedrock can be found. So the development of the rooting depth should be similar as presented in the arid region.

Technical note:

Legend Figure 1 change contrained to constrained

Figure 7 the color codes of the arrows and lines need to be explained, either in the legend or the figure caption

Figure S20 the figure caption mentions relationship between fAPAR and climate and landscape positions but the legend says WTD, please clarify.

---

## Author Comment (AC1) · 25 Apr 2020

**Answer on Anonymous Referee #2**

Dear Reviewer,

Thank you for your review. Below we address the comments you made on the paper.

**Main points:**

- *"How have the landscape positions been validated, the map presented in Figure S6 makes sense at the global scale, but how valid are the results if you look at the landscape scale, where the authors develop their conceptual framework? I assume this can be easily done with global topography data such as SRTM. I would encourage the authors to discuss this a little bit more in detail since the landscape position is a critical part in your analysis."*

The landscape classification was based on the same water table depth dataset used for the correlation calculations. This water table depth map data was produced with global topography data as one of the input datasets. The resulting classification was validated visually against some geological literature on sample regions. As with any classification some locations will be misclassified, but it does pick up on even quite small landscape features, as I will demonstrate in two examples.

Example number one is a closeup of the Netherlands. The bigger landscape units, the 'wetland and open water' class, defines most of the west and north of the Netherlands, the areas with extensive lowland polders (Hartemink and Sonneveld, 2013). Smaller landscape units are visible as well; the coastal dunes in the west of the Netherlands and the individual push moraines in the Veluwe complex (central Netherlands) can be distinguished (for reference see Overmeeren, 1997). Also, smaller units are visible such as the river levees of both the Meuse and Waal (the last part of the Rhine).

[Figure]

*Figure 1: Landscape classification closeup from the Netherlands*

The second example shows northern Italy. The Alps show up as the highest locations in this example, with some detail in this mountainous area. The Apennines appear as mountainous but clearly lower than the Alps. In between, the Po valley shows up as low lying area and the delta (close to Venice) shows up as 'wetland and open water'. Smaller units are picked up as well; several lakes appear, most notably Lago di Garda, just under the Alpine region. Just below the Alps, the Euganean Hills – protrusions in the Po valley of volcanic origin – show up as well.

[Figure]

*Figure 2: Landscape classification closeup of northern Italy*

We decided to not include these visual examples in the paper for brevity. We will add a sentence to the paper describing that the classification has been validated based on visual inspection based on several sample regions.

- *"In the description of the Ecohydrological classes in section 2.3 I would stronger present the effects of temperature on forest growth in the higher landscape positions to avoid misinterpretations. Since this class is mostly present in the higher landscape classes of the temperate regions."*

Temperature plays indeed an important role in this story, especially in the mountainous areas. As mentioned by the reviewer the class "composite" is often found in these areas. In the class descriptions we explain this temperature effect; vegetation grows better in lower landscape positions. This can be attributed not only to water being more available in the lower lying areas, also temperature, soil depth and nutrient availability are more favorable for tree growth in these lower landscape positions. As this class is linked to many different factors we decided to call it 'composite'. We believe this addressed the issue of misinterpretation, but will put some more emphasis on temperature in the descriptions of the classes.

**Minor points**

- *"statement line 12 to 14 In my mind this statement is only true for water limited areas. For more humid, energy limited environments like the temperate and boreal zones I am not sure whether water availability determines whether vegetation grows or not especially when it comes to trees. I would argue that in the colder climates and higher mountainous areas plant growth an especially Tree growth is also limited by temperature which can be clearly also seen by the tree line distribution in the high mountain areas in the temperate regions as well as in the northern climates."*

Although we definitely agree with the more nuanced picture the reviewer describes, the point we wanted to make with this sentence is that by looking globally at the vegetation distribution water availability is key in understanding the patterns. If insufficient water is present, vegetation does not grow. On the other hand, if enough water is present it does not mean that vegetation definitely does have to grow. Besides temperature, also soil depth, stability and toxicity might be other factors preventing plants to grow at all. To avoid ambiguity, we will change the sentence to

"Water availability is a prerequisite for vegetation growth, while plants influence the local hydrological situation through interception of precipitation and transpiration of water absorbed in the root zone."

- *" Statement line 16 to 20 This statement might be true on large continental scale, however as experiences of the drought years 2018 and 2019 in Europe have shown that forests mainly consisting of species trees species with shallow roots such as spruces suffered serious damages during the droughts."*

This is indeed a good point. This statement is meant to address the point that trees have deeper roots than other vegetation and because they are long lived species they need to be adapted to the local climate and hydrological conditions. This makes them more resilient to weather anomalies (on an ecosystem level) but extremes can still be deadly, especially for varieties (or relatively young forests) with shallower root systems. The drought of 2018 and 2019 was quite extreme for the European climate. We will change the sentence to the following:

"Because they can take up water from considerable depth with their extensive root systems, trees are highly adapted to the local climate and hydrologic regime, making them *more* resilient to weather anomalies, such as prolonged periods of drought"

- *" Statement line 45 to 47 Rooting depth also depends soil properties like the existence of a layer of higher density in the soil profile. This is for instance very often the case in landscapes which have developed after the glaciation period or have been influenced by glaciation (e.g. in North America, Central Europa, Northern Part of Asia)."*

Thank you for the nuance, we will add the following statement to the manuscript:

"Exceptions can occur for various reasons, such as slope instability, insufficient soil depth and the presence of hardpans in the soil."

- *"Figure 6: I would have expected a stronger temperature effect on forest growth also in the lower landscape classes like low mountain areas and hilly landscapes. How can this be explained?"*

The effect of temperature in these areas is most likely present, but the effect of increased precipitation at the highest locations seems to be dominant. For the tropical climate (Af) this region in classified as 'Convergence dominated'. One component explaining higher growth in lower lying areas is most likely related to the access to the groundwater, but also temperature might play a role in this trend.

- *"Figure 7: For the boreal and temperate regions the figure indicates a deep and unchanging rooting depth from low mountainous, mountainous and high mountainous regions. This is misleading. In fact in these areas the rooting depth decreases with elevation. In the higher elevations only shallow soils over bedrock can be found. So the development of the rooting depth should be similar as presented in the arid region.*

Good point, we will adjust the figure

- *"Legend Figure 1 change contrained to constrained"*

We will change this

- *" Figure 7 the color codes of the arrows and lines need to be explained, either in the legend or the figure caption"*

The colors are directly linked to the colors of figure 1, we will accentuate this link in the caption. The direction of the arrows corresponds to the correlation between WTD and fPAR, as obtained from figure 1, we will also clarify this.

- *"Figure S20 the figure caption mentions relationship between fAPAR and climate and landscape positions but the legend says WTD, please clarify."*

Thank you for spotting the mistake, we will change to caption.

**References**

Hartemink, A. E., & Sonneveld, M. P. W. (2013). Soil maps of The Netherlands. *Geoderma*, *204–205*, 1–9. https://doi.org/10.1016/j.geoderma.2013.03.022

Van Overmeeren, R. A. (1998). Radar facies of unconsolidated sediments in The Netherlands: A radar stratigraphy interpretation method for hydrogeology. *Journal of Applied Geophysics*, *40*(1–3), 1–18. https://doi.org/10.1016/S0926-9851(97)00033-5

---

## Author Response (AR1)

Dear Editor,

Hereby we submit the revised version of the manuscript entitled "Global distribution of hydrological controls on forest growth". We received positive and constructive feedback during the review process leading to two major changes in the manuscript: precipitation was replaced by aridity, to include thermal control on hydrology through potential evapotranspiration (PET), as suggested by both reviewers and yourself. Secondly, we updated the methodology of calculating significance threshold slightly and compared the resulting classification with two different procedures and found the results to overlap in over 90% of the pixels. This strong overlap gave us confidence in the methodology, and we believe that the results and conclusions are robust. The reviewers seem to further agree that the manuscript is interesting and well written. We hope that the rebuttal will be positively received by the reviewers.

Best regards, on behalf of all co-authors,

Caspar Roebroek,

Utrecht, 22-6-202

**Anonymous Referee #1**

In this interesting work, the authors classify the land surface into patterns of dominant effects on forest growth by precipitation or water table depth (wtd) using satellite imagery of fPAR as a proxy for forest growth, modelled wtd, and measured and interpolated precipitation. The analysis is based on spatial correlations between long term averages of the high resolution data sets. They find that the relationship between precipitation and fPAR is prevalent, but the effects of wtd are still widespread and important. The authors also illustrate variations of the relationships as the result of local climate conditions and landscape characteristics. As the authors convincingly explain in a paragraph in the discussion, the results prove that in current modelling approaches of the land surface using exclusively precipitation as a hydrologic control on forest growth is not sufficient and the work is therefore timely and relevant. The paper is very well written, the analysis steps are clearly explained including underlying assumptions and the results are logically structured and interpreted.

I was wondering, however, why the authors used full correlations all the way through their analysis when the scope was to actually isolate the hydrological control/ contribution. As mentioned in the description of several ecohydrological classes and sometimes in the interpretation of the results, spatial covariates like temperature play a role and will explain some of the patterns of correlations found, especially those with precipitation. So, why not remove at least the contribution of spatial gradients in temperature as a known important control on forest growth by partial correlations to narrow down the contributions of the hydrological controls?

We agree with the importance of discussing temperature as a major contributor to determining vegetation growth. However, it should be noted that temperature effects alone cannot easily be removed due to their strong covariability with P. In order to account for the effect of temperature, we therefore took a different approach. We replaced precipitation with aridity (P/PET) in the analysis procedure to express climate driven water availability rather than only water supply. Interestingly, the results and main conclusions remained almost identical, strengthening the belief that energy and water availability are inherently intertwined and in most instances inseparable in such large-scale data analyses. By introducing temperature as a covariable, we could make the descriptions of high altitude and cold climates more straightforward, thus strengthening the conclusions.

I might pose a similar question regarding the relationship between precip and wtd, which might also be split more rigorously. However, the authors take this into account in the interpretation and explain well in the paper, so I do not pose this a major point of discussion.

We included a global map representing the correlation between P/PET and WTD (in the supplementary material) and stressed their link in the class descriptions to clarify this point better. Especially the classes on the diagonal of Figure 2 (oxygen stress, neutral, and water limited) are strongly dependent on the link between climate and landscape driven water availability, and represent the classical view of global vegetation growth assessment (such as Köppen-Geiger climate classification). One of our main results is that by separating the

gradients, most cases are not represented in this diagonal, which caused us to conclude that on global scale, landscape changes climate driven expectations on vegetation growth substantially.

Overall, the work the authors present in their paper is scientifically interesting and relevant, methodologically mostly logical (next to the one major point stated above, I pose some minor methodological questions below that need clarification or justification in my opinion), and is presented in an excellent way regarding both text and figures. I see the need for revision and minor clarifications before publication.

Minor aspects that need clarification/ discussion and potentially changes in the manuscript:

- Consistency of the long-term averages of the data sets: As shown in table 1 of the main text, the length and the periods that they represent differ by 10 years and more between individual data sets. How might this affect the consistency of the long-term averages that are the basis of the analysis?

In this study we are using data with high resolution, obtained from long term observations. The climate variables and water table depth are from an earlier period than fAPAR. We expect changes in climate to be more regional, and therefore not affect the gradient within each 15 x 15 window. Also, we do not expect WTD to have changed considerable, as the landscape forms are still the same. This will not always work, in such cases as local water extractions being implemented, but on average on the whole world these changes will be relatively minor. Therefore, the period mismatch will presumably lower the correlation values somewhat, but significance might still be tested. We believe this mismatch will not substantially influence the conclusions.

Secondly, the data availability of at least the fPAR dataset will vary seasonally due to snow or cloud effects. Has this issue been considered and taken into account in some way in order to prevent the longterm averages to be seasonally biased?

The following fPAR data source has been used: https://developers.google.com/earth-engine/datasets/catalog/MODIS_006_MCD15A3H. It uses the best value for each pixel in four consecutive days. The long-term averaging has been performed on this data directly to mostly avoid phenomena such as cloud effects.
The biases in the averaging and the seasonal cycle are addressed by the regional scale on which the correlations are calculated. Within each 15x15 grid cell window these biases are assumed homogeneous. The absolute values will shift, but the gradients would remain equal, yielding similar correlation values. We added this assumption to the methodology section to avoid ambiguity.

- Is the scope to analyse hydrological control of trees or of forests? From the title I expected only forests, but basically all results are based on any pixels having a canopy height>3m independent of any definition of a forest, eg tree density.

We are interested in all natural vegetation but applied a filter of 3 meters to exclude most farmland as the signal of the natural controls will be heavily distorted. This threshold to

classify trees seemed reasonable and is often used in literature. A pixel containing vegetation with an average height of 3 meters is consequently called forest. For clarity we will explain the use of 'forest' better in the methodology.

The assumption that the 'translation from fAPAR values to photosynthetic activity are homogeneous' (l. 91) in each moving window appears strong when only the threshold of 3m is used as a filter criterion and in reality several vegetation types might be mixed in the pixel. A slight rewording in the first and a clarification in the second case are appreciated.

This is indeed a good point. We assume not only vegetation type, but also vegetation age to be homogeneously distributed over each window (which would then make it justifiable to also assume a similar conversion function between fPAR and actual photosynthesis). Locally this assumption will not always hold, but we believe this assumption is reasonable for a global synthesis and that the errors in the assumption do not substantially alter the final observations and conclusions.

- Are only those correlations displayed and evaluated that were tested as significant? Have you tried whether the results strongly change of you apply other criteria in addition, such as a (higher) correlation threshold? A threshold of 0.11 for a significant correlation for fully available spatial windows (l.101) is quite low as to have a strong meaning for the interpretation.

We indeed only analysed and evaluated the significant correlation values. To strengthen the results and compensate for the spatial autocorrelation of the samples (see comment by reviewer #2) within windows we altered the methodology slightly and reduced the degrees of freedom used in the t-test, based on an adaptive approach which compares the t-test approach with a bootstrapping analysis and permutation test. Finally, we compared the three approaches directly by classifying South America in the ecohydrological classes. All methods show a very high degree of overlap, which makes us confident that the conclusions are robust and method independent. The results of this comparison are included in the supplementary information.

**Anonymous Referee #2**

General comments: In their work the authors present a method where they link forest growth patterns to precipitation and water table depth on the global scale. They use long term high resolution satellite products for fAPAR, modelled groundwater table depth and globally distributed precipitation. They have developed a classification scheme of ecohydrological classes based on the correlation between water table depth and fAPAR as well as precipitation and fAPAR. To assess the impact of climate and landscape position on the distribution of these ecohydrological classes, the authors make use of the Köppen-Geiger classification (climate) and 7 landscape classes derived from the global water table depth map. They discuss and illustrate their findings for several regions of the globe. In the end, based on their findings they develop a conceptual framework of forest growth and its link to hydrologic gradients in the landscape. Finally, the authors discuss how their findings can support a better representation of forest growth in global environmental modelling which is still a relevant question. The manuscript is well written and conceivable. It is provided with an extensive supplement which contributes to the understanding of the manuscript. There are some small points which should be clarified because they might lead to misinterpretations which, I will address in specific comments section and section for technical correction. However, I have two major points which should be discussed by the authors with more emphasis.

How have the landscape positions been validated, the map presented in Figure S6 makes sense at the global scale, but how valid are the results if you look at the landscape scale, where the authors develop their conceptual framework? I assume this can be easily done with global topography data such as SRTM. I would encourage the authors to discuss this a little bit more in detail since the landscape position is a critical part in your analysis.

The landscape classification was based on the same water table depth dataset used for the correlation calculations. This water table depth map data was produced with global topography data as one of the input datasets. The resulting classification was validated visually against some geological literature on sample regions. As with any classification some locations will be misclassified, but it does pick up on even quite small landscape features, as I will demonstrate in two examples.

Example number one is a closeup of the Netherlands (Figure 1). The bigger landscape units, the 'wetland and open water' class, defines most of the west and north of the Netherlands, the areas with extensive lowland polders (Hartemink and Sonneveld, 2013). Smaller landscape units are visible as well; the coastal dunes in the west of the Netherlands and the individual push moraines in the Veluwe complex (central Netherlands) can be distinguished (for reference see Overmeeren, 1997). Also, smaller units are visible such as the river levees of both the Meuse and Waal (the last part of the Rhine).

[Figure]

*Figure 1: Landscape classification closeup from the Netherlands*

The second example shows northern Italy (Figure 2). The Alps show up as the highest locations in this example, with some detail in this mountainous area. The Apennines appear as mountainous but clearly lower than the Alps. In between, the Po valley shows up as low lying area and the delta (close to Venice) shows up as 'wetland and open water'. Smaller units are picked up as well; several lakes appear, most notably Lago di Garda, just under the Alpine region. Just below the Alps, the Euganean Hills – protrusions in the Po valley of volcanic origin – show up as well.

[Figure]

*Figure 2: Landscape classification closeup of northern Italy*

We included these examples in the supplementary material. Additionally, we will add a sentence to the main text describing that the classification has been validated based on visual inspection based on several sample regions.

In the description of the Ecohydrological classes in section 2.3 I would stronger present the effects of temperature on forest growth in the higher landscape positions to avoid misinterpretations. Since this class is mostly present in the higher landscape classes of the temperate regions.

Based on the comments of reviewer 1 and the editor we decided to more thoroughly include the effect of temperature by substituting precipitation with aridity (P/PET). For details see above and in the updated manuscript.

Specific comments

statement line 12 to 14 In my mind this statement is only true for water limited areas. For more humid, energy limited environments like the temperate and boreal zones I am not sure whether water availability determines whether vegetation grows or not especially when it comes to trees. I would argue that in the colder climates and higher mountainous areas plant growth an especially Tree growth is also limited by temperature which can be clearly also seen by the tree line distribution in the high mountain areas in the temperate regions as well as in the northern climates.

Although we definitely agree with the more nuanced picture the reviewer describes, the point we wanted to make with this sentence is that by looking globally at the vegetation distribution, water availability is key in understanding the patterns. If insufficient water is present, vegetation does not grow. On the other hand, if enough water is present it does not mean that vegetation definitely does have to grow. Besides temperature, also soil depth, stability and toxicity might be other factors preventing plants to grow at all. To avoid ambiguity, we will change the sentence to:

"Water availability is a prerequisite for vegetation growth, while plants influence the local hydrological situation through interception of precipitation and transpiration of water absorbed in the root zone."

Statement line 16 to 20 This statement might be true on large continental scale, however as experiences of the drought years 2018 and 2019 in Europe have shown that forests mainly consisting of species trees species with shallow roots such as spruces suffered serious damages during the droughts.

This is indeed a good point. This statement is meant to address the point that trees have deeper roots than other vegetation and because they are long lived species they need to be adapted to the local climate and hydrological conditions. This makes them more resilient to weather anomalies (on an ecosystem level) but extremes can still be deadly, especially for varieties (or relatively young forests) with shallower root systems. The drought of 2018 and 2019 was quite extreme for the European climate. We will change the sentence to the following:

"Because they can take up water from considerable depth with their extensive root systems, trees are highly adapted to the local climate and hydrologic regime, making them *more* resilient to weather anomalies, such as prolonged periods of drought"

Statement line 45 to 47 Rooting depth also depends soil properties like the existence of a layer of higher density in the soil profile. This is for instance very often the case in landscapes which have developed after the glaciation period or have been influenced by glaciation (e.g. in North America, Central Europa, Northern Part of Asia).

Thank you for the nuance, we will add the following statement to the manuscript:

"Exceptions can occur for various reasons, such as slope instability, insufficient soil depth and the presence of hardpans in the soil."

Figure 6: I would have expected a stronger temperature effect on forest growth also in the lower landscape classes like low mountain areas and hilly landscapes. How can this be explained?

The effect of temperature in these areas is most likely present, but the effect of increased precipitation at the highest locations seems to be dominant.

Figure 7: For the boreal and temperate regions the figure indicates a deep and unchanging rooting depth from low mountainous, mountainous and high mountainous regions. This is misleading. In fact in these areas the rooting depth decreases with elevation. In the higher elevations only shallow soils over bedrock can be found. So the development of the rooting depth should be similar as presented in the arid region.

Corrected

Technical note:

Legend Figure 1 change contrained to constrained

Corrected

Figure 7 the color codes of the arrows and lines need to be explained, either in the legend or the figure caption

Corrected

Figure S20 the figure caption mentions relationship between fAPAR and climate and landscape positions but the legend says WTD, please clarify.

Corrected

[revised manuscript text omitted]

---

## Author Response (AR2)

Dear Editor,

Hereby we submit the revised version of the manuscript "Global distribution of hydrological controls on forest growth". The main difference with the previous version is the use of the term "humidity" rather than "aridity". Additionally, some minor textual changes are made to more precisely reflect the concept of the humidity index. We hope these changes are positively received.

Best regards, on behalf of all co-authors,

Caspar Roebroek,

Utrecht, The Netherlands
13-8-2020

[revised manuscript text omitted]